# Reductions in commuting mobility correlate with geographic differences in SARS-CoV-2 prevalence in New York City

Stephen M. Kissler [1,11], Nishant Kishore [2,11], Malavika Prabhu[3,11], Dena Goffman[4,11], Yaakov Beilin[5,6,11], Ruth Landau[7], Cynthia Gyamfi-Bannerman[4], Brian T. Bateman[8], Jon Snyder[3], Armin S. Razavi [3], Daniel Katz[5,6], Jonathan Gal[5], Angela Bianco[6], Joanne Stone[6], Daniel Larremore [9,10], Caroline O. Buckee [2] & Yonatan H. Grad [1✉]

SARS-CoV-2-related mortality and hospitalizations differ substantially between New York City neighborhoods. Mitigation efforts require knowing the extent to which these disparities reflect differences in prevalence and understanding the associated drivers. Here, we report the prevalence of SARS-CoV-2 in New York City boroughs inferred using tests administered to 1,746 pregnant women hospitalized for delivery between March 22nd and May 3rd, 2020. We also assess the relationship between prevalence and commuting-style movements into and out of each borough. Prevalence ranged from 11.3% (95% credible interval [8.9%, 13.9%]) in Manhattan to 26.0% (15.3%, 38.9%) in South Queens, with an estimated city-wide prevalence of 15.6% (13.9%, 17.4%). Prevalence was lowest in boroughs with the greatest reductions in morning movements out of and evening movements into the borough (Pearson R = −0.88 [−0.52, −0.99]). Widespread testing is needed to further specify disparities in prevalence and assess the risk of future outbreaks.

[1] Department of Immunology and Infectious Diseases, Harvard T.H. Chan School of Public Health, Boston, MA, USA. [2] Department of Epidemiology, Harvard T.H. Chan School of Public Health, Boston, MA, USA. [3] Department of Obstetrics and Gynecology, Weill Cornell Medicine, New York, NY, USA. [4] Department of Obstetrics and Gynecology, Columbia University Irving Medical Center, New York, NY, USA. [5] Department of Anesthesiology, Perioperative and Pain Medicine, Icahn School of Medicine at Mount Sinai, New York, NY, USA. [6] Department of Obstetrics Gynecology and Reproductive Sciences, Icahn School of Medicine at Mount Sinai, New York, NY, USA. [7] Department of Anesthesiology, Columbia University Irving Medical Center, New York, NY, USA. [8] Department of Anesthesiology, Brigham and Women's Hospital, Boston, MA, USA. [9] Department of Computer Science, University of Colorado Boulder, Boulder, CO, USA. [10] BioFrontiers Institute, University of Colorado Boulder, Boulder, CO, USA. [11] These authors contributed equally: Stephen M. Kissler, Nishant Kishore, Malavika Prabhu, Dena Goffman, Yaakov Beilin. ✉email: ygrad@hsph.harvard.edu

Pronounced geographic differences in hospitalization and mortality rates have emerged as a hallmark of the ongoing SARS-CoV-2 pandemic. In New York City, the epicenter of the SARS-CoV-2 epidemic in the United States, deaths and hospitalizations per capita due to COVID-19 were nearly twice as high in the Bronx as in neighboring Manhattan as of 25 April 2020[1]. To the extent that this variation reflects the cumulative incidence of SARS-CoV-2 infections, there may be wide disparities in exposure to the novel coronavirus across New York City. Furthermore, if exposure leads to protection from re-infection, substantial levels of immunity may have already accrued in some areas of the city, while other neighborhoods may still be susceptible to a major outbreak. Understanding this risk landscape is key for informing plans to responsibly resume commerce in the coming months.

The local prevalence of SARS-CoV-2 infection depends on a number of factors, including the patterns of contacts among people within and between communities. Physical distancing interventions, including the "New York State on PAUSE" executive order starting 22 March[2], have dramatically changed the behaviors that drive these contacts. COVID-19 hospitalization and mortality rates are an imperfect proxy of prevalence, since these measures also depend on access to care, age, social determinants of health, and the rates of underlying medical conditions as well as non-disease-related phenomena such as hospital overload. Measuring the prevalence of SARS-CoV-2 infection has been difficult because tests are generally only administered for patients with presumed COVID-19 illness, leaving mild, asymptomatic, and presymptomatic cases uncounted. Imperfect test sensitivity adds an additional layer of complexity to extrapolating the results of SARS-CoV-2 tests to the general population.

Here, we report that SARS-CoV-2 prevalence varied substantially between New York City boroughs between 22 March and 3 May 2020. These differences in prevalence correlate with antecedent reductions in commuting-style mobility between the boroughs. Our findings are based on quantitative polymerase-chain-reaction SARS-CoV-2 tests administered to 1746 women hospitalized for delivery and aggregated mobility data from the Facebook Data for Good initiative. These findings underscore the need to support neighborhoods unable to fully comply with social distancing interventions with enhanced contact tracing, protective equipment, and other interventions aimed at reducing transmission.

## Results

**SARS-CoV-2 prevalence varies between New York City boroughs**. To estimate SARS-CoV-2 prevalence by New York City borough, we analyzed SARS-CoV-2 test results administered universally with informed consent to 2011 pregnant women admitted for delivery at four NewYork—Presbyterian Hospital campuses (Columbia University Irving Medical Center/NYP—CUIMC, Weill Cornell Medical Center/NYP—WCM, Lower Manhattan Hospital/NYP—LMH, and Queens Hospital/NYP—Queens), Mount Sinai Hospital (MSH), and Mount Sinai West (MSW) Hospital between 22 March and 3 May 2020. NYP—CUIMC tests included those from NYP—Morgan Stanley Children's Hospital and NYP-Allen Hospital. We excluded tests from women with a ZIP code outside of New York City ($n = 251$) or in Staten Island ($n = 14$) due to the small sample size from that borough, leaving tests from 1746 women (Table 1). Consistent with a recent report[3], 244 (14.0%) of the women tested positive for SARS-CoV-2. Of these, 55 (22.5%) reported symptoms including fever, cough, sore throat, chills, malaise, chest pain, shortness of breath, anosmia, or hyposmia. We combined these data with high-volume mobility data[4] from Facebook users

**Table 1 Characteristics of the study population.**

| Category | N | % |
|---|---|---|
| Total | 1746 | 100 |
| Site | | |
| NYP—CUIMC | 385 | 22.1 |
| NYP—LMH | 137 | 7.9 |
| NYP—Queens | 178 | 10.2 |
| NYP—WCM | 290 | 16.6 |
| MSH | 428 | 24.5 |
| MSW | 328 | 18.8 |
| SARS-CoV-2 test result | | |
| Positive | 244 | 14.0 |
| Negative | 1502 | 86.0 |
| Borough | | |
| Bronx | 309 | 17.7 |
| Brooklyn | 386 | 22.1 |
| Manhattan | 718 | 41.1 |
| North Queens | 275 | 15.8 |
| South Queens | 58 | 3.3 |
| Age | | |
| 15–19 | 21 | 1.2 |
| 20–24 | 167 | 9.6 |
| 25–29 | 346 | 19.8 |
| 30–34 | 588 | 33.7 |
| 35–39 | 470 | 26.9 |
| 40–44 | 139 | 8.0 |
| 45–49 | 13 | 0.7 |
| 50–54 | 2 | 0.1 |

*NYP—CUIMC New York Presbyterian Columbia University Irving Medical Center, NYP—WCM Weill Cornell Medical Center, NYP—LMH Lower Manhattan Hospital, NYP—Queens Queens Hospital, MSH Mount Sinai Hospital, MSW Mount Sinai West.*

capturing the number of daily trips made into and out of each borough to assess how changes in individuals' movement patterns may have contributed to geographic variation in SARS-CoV-2 prevalence.

Each SARS-CoV-2 test record was assigned to a borough on the basis of the three-digit prefix of the patient's ZIP code (Supplementary Table 1)[5]. To improve the spatial resolution, we separated Queens, the largest borough by land area, into North and South regions, delineated by the New York State Department of Health's neighborhood designations of North/Northeast/Northwest/West/West Central/Central and Jamaica/Rockaways/Southeast/Southwest, respectively (Supplementary Table 1)[5]. The percentage of tests positive for SARS-CoV-2 ranged from 10.0% (72/718) in Manhattan to 22.4% (13/58) in South Queens (Supplementary Table 2). We used a statistical framework[6,7] to estimate the population prevalence of SARS-CoV-2 infection by borough accounting for imperfect test sensitivity, which has been reported as low as 70%[8] (Fig. 1a, Supplementary Table 3). This framework estimates a Bayesian posterior distribution for the prevalence of infection in a population accounting for the size of the sample and imperfect test sensitivity. For example, a naïve estimate of the population prevalence given a sample of 10 positive and 90 negative tests might be 10%, but if the test has imperfect sensitivity, one would expect some of the 90 negative tests to actually be COVID-19 positive. The estimate of population prevalence needs to be adjusted upward accordingly. Conservatively estimating a test sensitivity of 90%, the mean estimated population prevalence of SARS-CoV-2 infection in Manhattan (11.3%, 95% credible interval (CI) [8.9%, 13.9%]) was substantially lower than in the Bronx (20.8%, [16.2%, 25.7%]) and South Queens (26.0%, [15.3%, 38.9%]) during the study period. Differences were not affected by assumed 80 and 70% sensitivity (Supplementary Table 3).

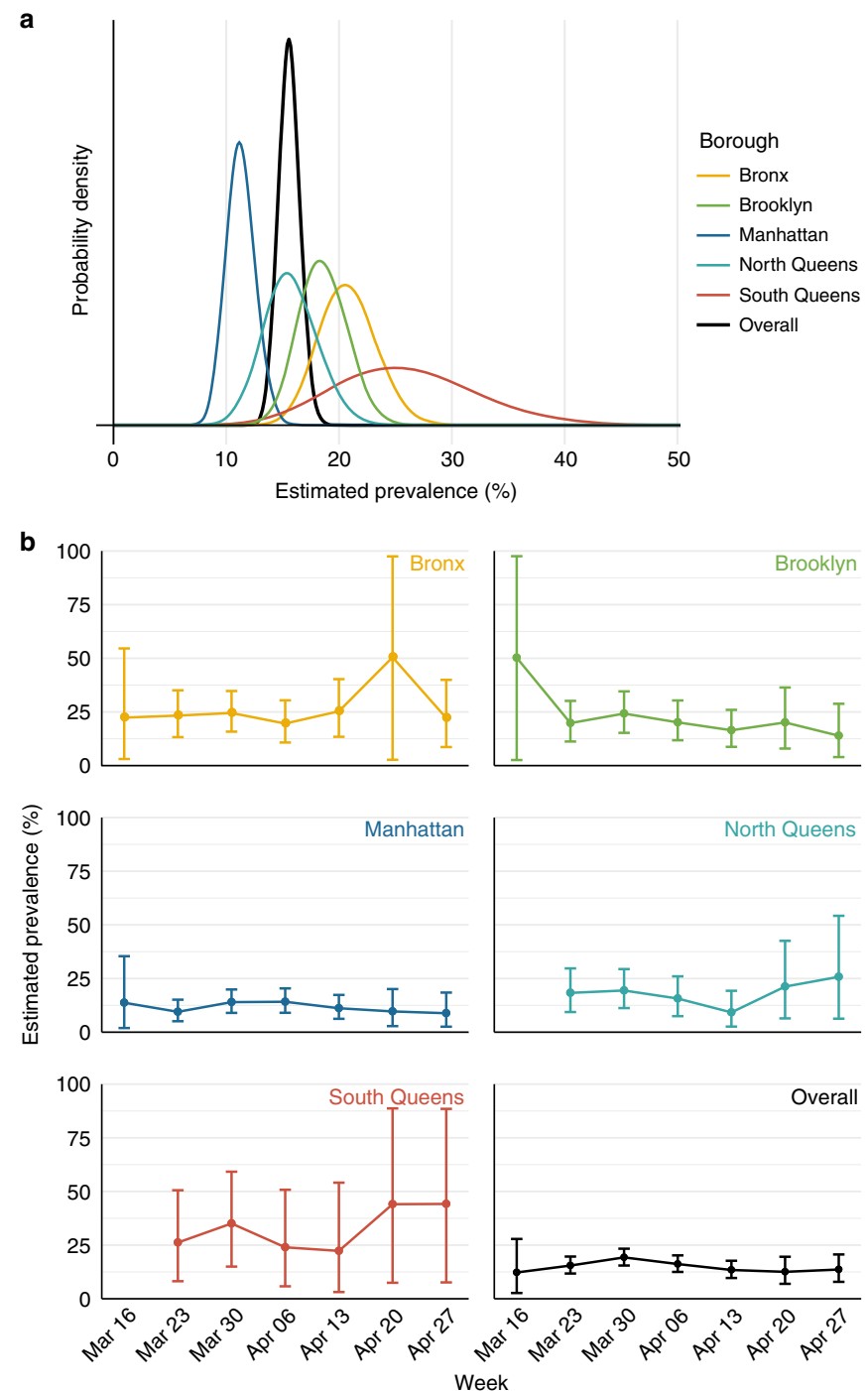

**Fig. 1 Posterior prevalence of SARS-CoV-2 infection by New York City borough. a** Posterior distribution of SARS-CoV-2 prevalence by borough (colors) and overall (black) across the study period. **b** Weekly mean posterior prevalence of SARS-CoV-2 infection by borough with 95% credible intervals. For both panels, the test was assumed to have perfect specificity and 90% sensitivity. There were no recorded SARS-CoV-2 tests from patients with Queens ZIP codes during the week of 16 March. Source data, including sample sizes by week and borough, are listed in Supplementary Table 3 and provided as a Source Data file.

Estimating the mean prevalence of SARS-CoV-2 infection in New York City using the data aggregated across all boroughs (15.6%, [13.9%, 17.4%]; Fig. 1a, black line) would mask these substantial geographic differences. A frequentist analysis of the same data confirms a significant difference in prevalence between boroughs (Chi Sq. test, $p = 0.00048$). The estimated prevalence of infection remained roughly constant over time (Fig. 1b) within statistical uncertainty (Chi Sq. test, $p = 0.29$), though the trends hint that

prevalence in the city as a whole may have risen until the week of 30 March and then tapered and leveled.

**SARS-CoV-2 prevalence correlates with reduced mobility.** To assess the possible relationship between variable reductions in between-borough movements and the subsequent prevalence of infection, we used mobility data provided by Facebook's Data for

Good program[4]. The data represent a cohort of ~1 million Facebook users in the New York City area who have location services enabled on their mobile device. The data provide 8-h snapshots of the number of transitions that occurred between ~1.2-km$^2$ patches in New York City. A transition is defined as a directional vector starting at the location where an individual spent the majority of their time during the preceding 8-h window of time and ending at the location where the same individual spent a majority of their time during the current 8-h window of time. We aggregated these data by borough and time of day (morning vs. evening, or 4 a.m. to 12 p.m. vs. 12 p.m. to 8 p.m.) and calculated the number of morning transitions out of each borough and evening transitions into each borough during the study period (22 March through 3 May) to approximate work-related commuting. We compared these values to the number of analogous transitions that occurred during the 45-day period preceding 26 February 2020, conditional on the day of the week and time of day. We chose to assess commuting between boroughs as opposed to within-borough movements because movements within a borough or neighborhood could include a variety of activities consistent with social distancing, whereas commuting between boroughs is likely to be associated with work and is therefore likely to be a good indicator of an inability to engage in social distancing. Furthermore, we chose to assess changes in movements rather than the absolute number of trips since the

data only capture a person's "modal" location (the location where they spent the most time during an 8-h interval), making the data better suited to summarizing changes in bulk movements rather than fine-scale interpersonal mixing patterns. The magnitude of the reduction in commuting movements between boroughs ranged from 41.4% in South Queens to 68.7% in Manhattan (Supplementary Table 4). The mean estimated prevalence of SARS-CoV-2 infection by borough was strongly inversely correlated with the reduction in commuting movements (Pearson $R = -0.88$, $[-0.52, -0.99]$) in each borough (Fig. 2). The relationship was similar with commuting movements in the reverse direction, i.e., the number of movements into each borough in the morning and out of each borough in the evening (Supplementary Fig. 3). The relationship between prevalence and changes in within-borough movements was not significant, as hypothesized (Supplementary Fig. 1, Supplementary Fig. 2). The mean and 95% CI for the Pearson correlation coefficient were calculated by drawing from each borough's posterior prevalence distribution (Fig. 1a) 10,000 times and recalculating the coefficient for each set. In a sensitivity analysis, we assessed the relationship between estimated SARS-CoV-2 prevalence and different metrics of movement including within-borough movement (Supplementary Fig. 1), total movement (Supplementary Fig. 2), reverse commuting-type movements (Supplementary Fig. 3), and total movements in/out of each borough (Supplementary Fig. 4). All

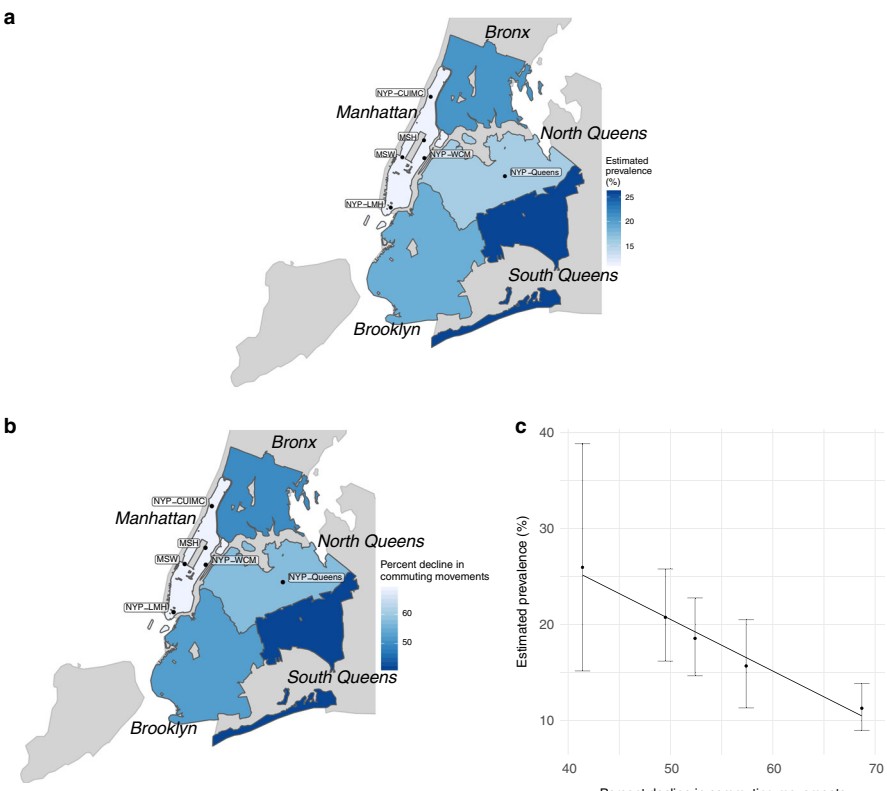

**Fig. 2 SARS-CoV-2 prevalence is lower in boroughs with greater declines in commuting movements. a** Estimated mean prevalence of SARS-CoV-2 infection by borough assuming a test with perfect specificity and 90% sensitivity. **b** Percent decline in commuting movements by borough during the study period compared to the 45 days preceding 26 Feb 2020. Commuting is measured as the total number of morning transits out of each borough and evening transits into each borough. Note the reverse scale, so that deep blue corresponds to higher prevalence in **a** and to a smaller decline in commuting in **b. c** Relationship between estimated prevalence of SARS-CoV-2 infection and decline in commuting movements by borough (Pearson $R = -0.88$, $[-0.52, -0.99]$). Points depict the mean posterior prevalence for each borough as a function of percent decline in commuting movements and error bars represent 95% credible intervals for the posterior prevalence distributions. Sample sizes are as follows: Bronx, $n = 309$; Brooklyn, $n = 386$; Manhattan, $n = 718$; North Queens, $n = 275$; South Queens, $n = 58$ tests. New York NYP—CUIMC Presbyterian Columbia University Irving Medical Center, NYP—WCM Weill Cornell Medical Center, NYP—LMH Lower Manhattan Hospital, NYP—Queens Queens Hospital, MSH Mount Sinai Hospital, MSW Mount Sinai West. Source data are listed in Supplementary Tables 3 and 4, and provided as a Source Data file.

relationships were weaker than the relationship between estimated prevalence and commuting-type movements.

## Discussion

The prevalence of SARS-CoV-2 infection between 22 March and 3 May 2020 differed substantially between New York City boroughs and was related to reductions in daytime commuting-style movements into and out of each borough relative to the previous 2 months. The estimated prevalence in Manhattan was substantially lower than in Queens and the Bronx, consistent with geographic differences in cumulative hospitalizations and mortality[1]. The variations in mobility across neighborhoods likely depended on factors including the distribution of essential workers and of resources to support distancing. If the differences in prevalence correlate with differences in population immunity, Manhattan may remain at higher risk of a major resurgence than the other boroughs as social distancing measures are relaxed, particularly when people who have left the city during the lockdown return.

Our findings are subject to a number of limitations. Women hospitalized for delivery may not be representative of the population[9,10]. Pregnancy may also dampen the immune response to the virus[11], possibly leading to a different duration of infection and therefore a biased representation of SARS-CoV-2 infection in pregnant women vs. the rest of the population. We have used mobility data as a proxy for physical distancing, but the mobility data do not perfectly capture the interpersonal contacts that underlie the transmission of SARS-CoV-2, nor do they necessarily capture the demographics of the women tested here for SARS-CoV-2. A direct causal link between physical distancing and the reduction in transmission cannot be drawn, because the ability to physically distance may also be related to age, income, type of employment, type of housing, and other factors that could independently modulate risk of infection. In addition, just as the prevalence of infection in the boroughs is more heterogeneous than the aggregate prevalence across New York City would suggest, there may be substantial geographic heterogeneity in prevalence within boroughs that is not captured in our study.

In conclusion, mobility patterns consistent with commuting correlate with the prevalence of SARS-CoV-2 infection in New York City boroughs. Large parts of the city may remain at risk for substantial SARS-CoV-2 outbreaks. These results highlight the need to provide greater support to neighborhoods unable to comply with social distancing interventions and that widespread SARS-CoV-2 testing remains key for assessing geographic disparities in infection prevalence, allowing for more tailored interventions and a better assessment of the risk of additional outbreaks.

## Methods

**Study design**. Quantitative polymerase-chain-reaction SARS-CoV-2 tests were administered uniformly to 2011 women hospitalized for delivery between 22 March and 3 May or some subset thereof (see Error! Reference source not found.) at Columbia University Irving Medical Center/NYP—CUIMC, Weill Cornell Medical Center/NYP—WCM, Lower Manhattan Hospital/NYP—LMH, and Queens Hospital/NYP—Queens, MSH, and MSW hospital. Clinical samples were obtained using a nasopharyngeal swab, which was performed upon admission. All women were also screened for symptoms consistent with COVID-19.

**Mobility data**. The percent change in commuting-style movements between boroughs was assessed using data provided by Facebook's Data for Good program[4]. The data use agreement permits the analysis and representation of changes in movements over time but not raw numbers of movements. The data represent a cohort of ~1 million Facebook users in the New York City area who have location services enabled on their mobile device. The data provide 8-h snapshots of the number of transitions that occurred between ~1.2-km² patches in New York City. A transition is defined as a directional vector starting at the location where an individual spent the majority of their time during the preceding 8-h window of

time and ending at the location where the same individual spent a majority of their time during the current 8-h window of time. For each borough and period of the day (morning: 4 a.m. to 12 p.m.; evening: 12 p.m. to 8 p.m.; night: 8 p.m. to 4 a.m.), we calculated the number of transitions out of and into each borough during the study period (22 March through 3 May). We compared these values to the number of analogous transitions that occurred during the 45-day period preceding 26 February 2020, conditional on the day of the week and time of day. The changes in mobility are calculated with respect to the same cohort during the pre-pandemic and pandemic periods, so that changes in the population sizes of the boroughs (e.g., due to people leaving the city during the pandemic) are implicitly accounted for. For the main analysis, we restricted our attention to morning movements out of boroughs and evening movements into boroughs, which represent commuting-style movements.

**Statistical analysis**. The Bayesian posterior prevalence of SARS-CoV-2 infection in the population was estimated using a procedure developed to estimate infection prevalence from representative samples of limited size obtained using tests with imperfect sensitivity[6]. An online calculator is available[7]. The correlation between posterior population prevalence of infection and percent decline in mobility was computed using standard linear regression. The mean and 95% CI for the Pearson's correlation coefficient were calculated by sampling 10,000 times from the posterior prevalence distributions for each borough and recalculating the correlation. The analysis was conducted in R version 3.6.2[12]. Code is available at https://github.com/gradlab/COVID_NYC.

**Reporting summary**. Further information on research design is available in the Nature Research Reporting Summary linked to this article.

## Data availability

Source data are provided with this paper[13] and may be found at https://github.com/gradlab/COVID_NYC (https://doi.org/10.5281/zenodo.3967753). Testing data are also listed in Supplementary Table 2.

## Code availability

Code for this analysis is available at https://github.com/gradlab/COVID_NYC[13] (https://doi.org/10.5281/zenodo.3967753).

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

## Acknowledgements
We thank the women who contributed data to this study. We thank Matthew Sisson for assistance with data collection and Bailey Fosdick for helpful comments. The work was supported in part by the Morris-Singer Fund for the Center for Communicable Disease Dynamics at the Harvard T.H. Chan School of Public Health. The funding source had no role in study design; in the collection, analysis, and interpretation of data; in the writing of the report; and in the decision to submit the paper for publication.

## Author contributions
S.M.K: performed the analysis, wrote the manuscript; N.K.: performed the analysis, wrote the manuscript M.P.: acquired data, edited the manuscript; D.G.: acquired data, edited the manuscript; Y.B.: acquired data, edited the manuscript; R.L.: acquired data, edited the manuscript; C.G.-B.: acquired data, edited the manuscript; B.T.B.: acquired data, edited the manuscript; D.K.: acquired data, edited the manuscript; J.G.: acquired data, edited the manuscript; A.B.: acquired data, edited the manuscript; J.S.: acquired data, edited the manuscript; J.S.: acquired data, edited the manuscript; A.S.R.: acquired data, edited the manuscript; D.L.: assisted with the analysis, edited the manuscript; C.O.B.: conceived of the study, edited the manuscript; Y.H.G.: conceived of the study, edited the manuscript, oversaw the work.

## Competing interests
The authors declare no competing interests.

## Ethics approval
The study was deemed exempt (IRB20-0669) by the Harvard T.H. Chan School of Public Health Internal Review Board.
