## [Peer Review File · Nature Communications]

REVIEWER COMMENTS

Reviewer #1 (Remarks to the Author):

1. The description of the statistical methods used is extremely sparse (“We used a statistical framework to estimate...”). I recognize that this is likely due to space constraints and that the authors have provided appropriate citations which appear, at a glance, to explain the methodology in detail. Nonetheless, an extra 2-3 sentences here describing at a high level the statistical methods used would go a long way towards ensuring that readers understand how these data were analyzed, and the practical implications of the selected methodology.
2. When calculating the reduction in mobility, the authors state that the reference period is the 45 days before February 26th, 2020, but are not explicit about the post-epidemic-start period used for calculating this metric. I assume this is the same as the range over which the prevalence data were collected (ie, March 22nd-May 3rd), but in any case, it would be useful to state this explicitly.
3. The authors provide a reasonable justification for why they focus on between-borough movements only, ignoring within-borough movements. This does necessarily exclude a large number of commutes, however, and I wonder if it would be possible to include as a sensitivity analysis the correlations utilizing all of the movement data?
4. More explanation as to why the variable of interest with respect to mobility patterns is the reduction in mobility, as opposed to the absolute level of mobility post-implementation of physical distancing interventions, would be useful. Intuitively, the absolute level of mobility seems more directly related to transmission risk, than the relative change.
5. The analysis described here is relatively simple, but appropriate and interesting. Moreover, the authors are careful to qualify the results of this analysis, noting that this analysis does not provide support for a direct causal link between physical distancing and reduction in mobility. That said, I would suggest even more caution in this regard given the nature of the analysis (the main result is simply the correlation between two variables based on a sample size of 5)—for example, replacing “predict” with “are correlated with” in the title and conclusions.

Reviewer #2 (Remarks to the Author):

Kissler and Kishore et al. present a study investigating the relationship between the daily commuting volume and SARS-CoV-2 prevalence of New York City boroughs. To examine this relationship, the authors estimated each borough’s commuting patterns using Facebook mobility data and each borough’s SARS-CoV-2 prevalence by enrolling women hospitalized for delivery, who were screened for SARS-CoV-2 regardless of symptoms. The results suggest that boroughs with the largest reduction in daily commutes also reported the lowest SARS-CoV-2 prevalence. This is a timely, well designed, and well written study that provides important insight into the geographical disparities of both SARS-CoV-2 prevalence and the impact of social distancing measures. I would recommend this manuscript for publication after addressing a few questions.

1. What was the source for borough population size? Did the authors account for the change in population over time as residents left the city? If different proportions of residents from certain boroughs left, then it may be skewing the reduction in trips over time or the prevalence of cases.
2. Why did the study only focus on trip numbers as a function of the origin (i.e. trips leave origin borough in the morning and return to the borough in the evening)? Understanding the geospatial heterogeneity in the number and source of incoming commuters could also be informative. For instance, were there boroughs where residents significantly reduced their commuting patterns, but

commuters from other boroughs continued to travel to/from that borough each morning/evening? Similarly, could there be a correlation between a borough's change in incoming commuters and its SARs-CoV-2 prevalence?

3. I understand the authors focused on the 4am-12pm and 12pm-8pm time windows to capture typical morning/evening commute. Is it correct to assume that the number of inter-borough trips made between 8pm and 4am during these dates were insignificant? If so, maybe include a line saying that $< X\%$ of trips were made during these times or there was an insignificant change in trip numbers made during this time relative to non-COVID times, thus we focused on the 4-12 and 12-8 times.

4. Could the authors label the boroughs on the map in Figure 2?

5. Were there temporal differences in when each borough started to have significant changes in commuting volume? For instance, did the boroughs with a lower prevalence also the start to reduce commuting levels earlier? Alternatively, if all boroughs reduced commuting at the same rate at the same point in time, did the boroughs that started to re-open sooner end up having a higher prevalence by the end of the study period? Perhaps this might be addressed by including a column for weekly % change in commuting in Supplemental Table 3.

We thank the reviewers for their constructive comments. Our edits are marked in the new “tracked changes” version of the manuscript and our point-by-point responses are listed here in blue.

REVIEWER COMMENTS

Reviewer #1 (Remarks to the Author):

1. The description of the statistical methods used is extremely sparse (“We used a statistical framework to estimate...”). I recognize that this is likely due to space constraints and that the authors have provided appropriate citations which appear, at a glance, to explain the methodology in detail. Nonetheless, an extra 2-3 sentences here describing at a high level the statistical methods used would go a long way towards ensuring that readers understand how these data were analyzed, and the practical implications of the selected methodology.

Thank you; we have added such a description into Paragraph 4 of the main text.

2. When calculating the reduction in mobility, the authors state that the reference period is the 45 days before February 26th, 2020, but are not explicit about the post-epidemic-start period used for calculating this metric. I assume this is the same as the range over which the prevalence data were collected (ie, March 22nd-May 3rd), but in any case, it would be useful to state this explicitly.

This is correct; we now state this explicitly in Paragraph 5.

3. The authors provide a reasonable justification for why they focus on between-borough movements only, ignoring within-borough movements. This does necessarily exclude a large number of commutes, however, and I wonder if it would be possible to include as a sensitivity analysis the correlations utilizing all of the movement data?

We have included two supplemental figures to address this point depicting the relationship between SARS-CoV-2 prevalence and (a) the decline in within-borough movements (**Supplemental Figure 1**) and (b) the decline in all movements (**Supplemental Figure 2**). The relationships remain in the same direction, but the correlations are substantially weaker. In all boroughs except Manhattan, the amount of movement within the borough increased relative to the pre-pandemic baseline (see also the line graph at the end of this document). This increase in movements could come from two opposing mechanisms: either (a) a person has replaced trips out of the borough with trips within the borough (consistent with greater physical distancing), or (b) a person who normally remained stationary during the pre-pandemic period started moving more during the pandemic (consistent with less physical distancing). These effects cannot be disentangled and are likely the reason for the diluted correlations reported in these supplemental figures. Since changes in between-borough movements are more interpretable and less likely to be confounded by opposing types of distancing behaviors, we chose to report just those movements for our main analysis.

4. More explanation as to why the variable of interest with respect to mobility patterns is the reduction in mobility, as opposed to the absolute level of mobility post-implementation of

physical distancing interventions, would be useful. Intuitively, the absolute level of mobility seems more directly related to transmission risk, than the relative change.

We agree that an ideal mobility metric would be the total amount of movement (i.e. number of trips) that a person takes. The mobility data available from Facebook do not capture this level of detail; instead, they only capture the location at which a person spent most of their time during an 8-hour window. For example, the data cannot distinguish between a person who in the pre-pandemic period spent most of their time at home but took frequent brief trips to a grocery store, restaurant, bar, etc., but during the pandemic stayed exclusively at home. Because of this, bulk changes in long-distance movements appear to be a better measure of distancing given the limitations of the dataset. Further, the data license agreement with Facebook prohibits us from displaying or sharing raw movement numbers. We have added a sentence explaining these points in Paragraph 5.

5. The analysis described here is relatively simple, but appropriate and interesting. Moreover, the authors are careful to qualify the results of this analysis, noting that this analysis does not provide support for a direct causal link between physical distancing and reduction in mobility. That said, I would suggest even more caution in this regard given the nature of the analysis (the main result is simply the correlation between two variables based on a sample size of 5)—for example, replacing “predict” with “are correlated with” in the title and conclusions.

Thank you; we have made these changes.

Reviewer #2 (Remarks to the Author):

Kissler and Kishore et al. present a study investigating the relationship between the daily commuting volume and SARS-CoV-2 prevalence of New York City boroughs. To examine this relationship, the authors estimated each borough’s commuting patterns using Facebook mobility data and each borough’s SARS-CoV-2 prevalence by enrolling women hospitalized for delivery, who were screened for SARS-CoV-2 regardless of symptoms. The results suggest that boroughs with the largest reduction in daily commutes also reported the lowest SARS-CoV-2 prevalence. This is a timely, well designed, and well written study that provides important insight into the geographical disparities of both SARS-CoV-2 prevalence and the impact of social distancing measures. I would recommend this manuscript for publication after addressing a few questions.

1. What was the source for borough population size? Did the authors account for the change in population over time as residents left the city? If different proportions of residents from certain boroughs left, then it may be skewing the reduction in trips over time or the prevalence of cases. For the mobility analysis, the declines in mobility pertain to a fixed cohort of mobile phone users. We extracted data on the borough population sizes from the Facebook dataset; the population size remained fairly stable for all boroughs except Manhattan (see line graph at the end of this document). Since our metric for decline in mobility refers to a cohort of individuals, it may be interpreted as the percent reduction in mobility per person; that is, the metric is normalized by the population size and so inherently accounts for changes in the total population size.

The population size does not enter into the calculation of SARS-CoV-2 infection prevalence. The statistical inference procedure essentially asks what a limited sample can tell us about a population with 'infinite' size. As with any statistical estimation procedure, the generalizability depends on the representativeness of the sample. Our analysis cannot say anything about prevalence in the New Yorkers who left the city, but pertains only to the population that remained.

2. Why did the study only focus on trip numbers as a function of the origin (i.e. trips leave origin borough in the morning and return to the borough in the evening)? Understanding the geospatial heterogeneity in the number and source of incoming commuters could also be informative. For instance, were there boroughs where residents significantly reduced their commuting patterns, but commuters from other boroughs continued to travel to/from that borough each morning/evening? Similarly, could there be a correlation between a borough's change in incoming commuters and its SARS-CoV-2 prevalence?

We have addressed this sensitivity analysis in **Supplemental Figure 3** and associated edits in the main text. The relationship between prevalence and these 'reverse commuting-style movements' was similar to the relationship with the commuting-style movements assessed in the main analysis.

3. I understand the authors focused on the 4am-12pm and 12pm-8pm time windows to capture typical morning/evening commute. Is it correct to assume that the number of inter-borough trips made between 8pm and 4am during these dates were insignificant? If so, maybe include a line saying that < X% of trips were made during these times or there was an insignificant change in trip numbers made during this time relative to non-COVID times, thus we focused on the 4-12 and 12-8 times.

A substantial number of trips were also made into, out of, and within boroughs at night, so we chose to address this as an additional sensitivity analysis. **Supplemental Figure 4** depicts the relationship between declines in these types of movements and the estimated prevalence of SARS-CoV-2 infection. While declines in nighttime movements out and movements within boroughs are negatively correlated with the estimated prevalence, none of the correlations are significant at the $p = 0.05$ level.

4. Could the authors label the boroughs on the map in Figure 2?

Yes; this is done.

5. Were there temporal differences in when each borough started to have significant changes in commuting volume? For instance, did the boroughs with a lower prevalence also start to reduce commuting levels earlier? Alternatively, if all boroughs reduced commuting at the same rate at the same point in time, did the boroughs that started to re-open sooner end up having a higher prevalence by the end of the study period? Perhaps this might be addressed by including a column for weekly % change in commuting in Supplemental Table 3.

By the beginning of our study period, (March 22nd) all boroughs had already been placed under lockdown and movement had largely stabilized and remained consistent throughout the study

period (see line graph below). We do not have data from the transition period and so cannot assess whether the timing of changes in mobility affected prevalence in the boroughs.

REVIEWERS' COMMENTS:

Reviewer #1 (Remarks to the Author):

The authors have addressed my earlier comments. I have no further comments.

Reviewer #2 (Remarks to the Author):

The authors have addressed my questions and improved the manuscript with their revisions. I recommend this revised version for publication.

REVIEWERS' COMMENTS:

Reviewer #1 (Remarks to the Author):

The authors have addressed my earlier comments. I have no further comments.

Reviewer #2 (Remarks to the Author):

The authors have addressed my questions and improved the manuscript with their revisions. I recommend this revised version for publication.